# Comparison of Different Volatile Extraction Methods for the Identification of Fishy Off-Odor in Fish By-Products

**DOI:** 10.3390/molecules27196177

**Published:** 2022-09-21

**Authors:** Yuanyuan Zhang, Long Tang, Yu Zhang, Huanlu Song, Ali Raza, Wenqing Pan, Lin Gong, Can Jiang

**Affiliations:** 1Laboratory of Molecular Sensory Science, Beijing Technology and Business University, Beijing 100048, China; 2Hunan Province Jiapinjiawei Biotechnology Co., Ltd, Changde 415401, China; 3Wuzhou Testing Co., Ltd, Jining 273200, China

**Keywords:** fish scraps, fish soup, GC-O-MS, GC × GC-O-MS, AEDA, r-OAV

## Abstract

This study was conducted to analyze volatile odor compounds and key odor-active compounds in the fish soup using fish scarp and bone. Five extraction methods, including solid-phase microextraction (SPME), dynamic headspace sampling (DHS), solvent-assisted flavor evaporation (SAFE), stir bar sorptive extraction (SBSE), liquid-liquid extraction (LLE), were compared and SPME was finally selected as the best extraction method for further study. The volatile odor compounds were analyzed by gas chromatography-olfactometry-mass spectrometry (GC-O-MS) and comprehensive two-dimensional gas chromatography-olfactometry-mass spectrometry (GC × GC-O-MS) techniques, and the key odor-active compounds were identified via aroma extract dilution analysis (AEDA) and relative odor activity value (r-OAV) calculation. A total of 38 volatile compounds were identified by GC-O-MS, among which 10 were declared as odor-active compounds. Whereas 39 volatile compounds were identified by GC × GC-O-MS, among which 12 were declared as odor-active compounds. The study results revealed that 1-octen-3-one, 2-pentylfuran, (E)-2-octenal, 1-octen-3-one, hexanal, 1-octen-3-ol, 6-methylhept-5-en-2-one, (E,Z)-2,6-nondienal and 2-ethyl-3,5-dimethylpyrazine were the key odor-active compounds in the fish soup.

## 1. Introduction

China is a large producer of freshwater fish, and the production of fish products is steadily increasing year by year. China has been rich in fish products since ancient times, and Dongting Lake in Hunan Province is the main producer of freshwater fish, especially having large amounts of white chub [1]. During the processing of freshwater fish, especially in the processing of minced fish, large amounts of scraps such as fish heads, fish bones, fish tails, fish skins, and offal are produced [2]. Previously, it was found that the meat extraction rate of seven kinds of bulk freshwater fish, namely mackerel, grass carp, silver carp, bighead carp, crucian carp and bream, was the highest for mackerel (54.33%) and the lowest for bighead carp (32.80%), while the remaining 45% to 67% were fish head and fish bone scraps [3]. At present, these scraps are not utilized as value-added products, except a limited portion of them are used for processing feed fish meal, and the majority of them are directly discarded, which not only wastes a lot of resources but also pollutes the environment and increases the cost of environmental waste management [4]. An important issue faced by the aquatic food industry these days is the effective utilization of fish waste in order to generate economic benefits. The fish head is rich in nucleotides, amino acids and inorganic elements such as potassium, calcium, sodium and magnesium that contribute relatively to the flavor, and it is valuable to utilize it in seasoned soup, fish bone paste or aquatic seasoning base after proper treatment [5]. However, the removal of a fishy odor is a key step in the production of seasoning bases, which first requires the investigation of the nature of a fishy taste. In this study, fish scraps (fish heads and bones) were used as raw materials for boiling fish soup and then analyzing the off-flavor compounds in fish soup in order to provide a research background for the removal of fishy odor compounds as well as a scientific basis for studies such as the production of seasoning bases using fish scraps.

Recently, the odor compounds in fish have emerged as one of the most important studies for the development of fish seasonings, which are directly related to the sensory quality of the product. The fishy flavor is one of the main indicators used to determine the quality of fish flavoring, and it is formed by the joint action of a variety of odor compounds. Liu et al. [6] used SPME combined with the GC-MS technique to analyze fishy odor substances in the muscle (35 kinds), head (35 kinds) and skin (38 kinds) of tilapia, and nonanal, octanal and (E)-octenal were found to be the key fishy substances which were identified via AEDA analysis. Similarly, in the study of Li et al. [7], the DHS-GC-MS technique was employed to analyze the odor compounds produced during hydrolysis of grass carp, and 37 odor compounds were detected. In previous studies, a series of investigations were conducted on the volatile compounds of different varieties of fish, although the site, type and source of the study subject were not specified. However, these volatile compounds do not necessarily determine the key aroma compounds in the products. Therefore, the study of key odor-active compounds is also necessary. Although there have been some advances in the study of aroma-active substances in fish and boiled fish soup, most of the studies have been conducted with fish raw materials; it is interesting to think about the Maillard reaction, caramelization reaction and oxidation of fats and oils that may occur after heat treatment of foods, and these reactions produce various volatile compounds [8,9]; these types of reactions are also present in the boiling process of fish scraps, which helps to further investigate the odor compounds in them and provides a research basis for the preparation of seasonings using fish scraps.

GC-O-MS is the key approach to analyze odor compounds in fish soup. However, during the boiling process, the odor compounds in the fish soup may react in various ways, causing changes in the volatile components [10]. In this study, a new gas chromatographic technique, namely two-dimensional integrated gas chromatography-olfactometry-mass spectrometry (GC × GC-O-MS), was compared with conventional one-dimensional gas chromatography in order to analyze the sample; this two-dimensional gas chromatography allows the conversion between two modes (one- and two-dimensional analysis modes). The two-dimensional mode allows volatile compounds in the sample to be detected more quickly and efficiently [11]. With the continuous promotion of GC × GC-O-MS, this technique has been applied to the analysis of volatile odor compounds in various samples. For example, Yang et al. [12] used GC × GC-O-MS to identify the differences in volatile odorants of cinnamon tea leaves at different roasting temperatures and detected 97 aroma substances. Very recently, SPME combined with GC-O-MS and GC × GC-O-MS techniques were employed by Zhao et al. [13] to identify volatile odor compounds of pepper via AEDA and OAV analysis.

Therefore, in order to make the useful recovery of volatile compounds from fish by-products, the offal (fish head and bones) of silver carp from Dongting Lake, Hunan Province was utilized. The present study was mainly aimed to accomplish the following objectives: (1) to select the most suitable volatile extraction method from solid-phase microextraction (SPME), dynamic headspace sampling (DHS), solvent-assisted flavor evaporation (SAFE), stir bar sorptive extraction (SBSE) and liquid-liquid extraction (LLE); (2) the optimal extraction method from (1) was selected and combined with the aroma extraction dilution analysis to identify the key odor-active compounds of fish soup. The significance of this study includes the selection of an appropriate method to analyze the odor compounds in fish broth and to study the key odor compounds, in order to provide a research basis for futuristic studies dealing with the removal of fishy odor from fish soup, and to provide a theoretical basis for the production of flavors from fish broth boiled with fish scraps.

## 2. Materials and Methods

### 2.1. Samples and Chemicals

The fish heads and bones used in the study were supplied by Hunan Jiapinjiawei Biotechnology (Hunan, China). Firstly, 60 °C warm water was added to a heavy-bottomed pot having a fish head and bone; furthermore, the waste water was removed, and washing was performed. The samples were mixed with water in a ratio of 1:1 by volume, boiling was performed under high pressure at 121 °C for 2 h, and the dregs were filtered off to obtain fish soup. Finally, the samples were concentrated to about 25% of the total soluble solid content with a rotary evaporator at 55°C.

Ethyl ether, n-hexane, and anhydrous sodium sulfate, all having purities >99%, were purchased from Lab Gou e-mall (Beijing, China). 2-Methyl-3-heptanone and n-alkanes (C7–C30) were provided by Sigma-Aldrich (St. Louis, MO, USA.). Nitrogen gas (99.9992% purity) was obtained from Beijing AP BAIF Gases Industry Co., Ltd. (Beijing, China) and the liquid nitrogen was obtained from Xian Heyu Trading Co., Ltd. (Beijing, China).

### 2.2. Solid-phase Microextraction (SPME)

The SPME method was applied to extract the flavor compounds from fish soup according to the protocols mentioned by Li et al. [14]. Five grams of the sample and 0.3 μ L of 2-methyl-3-heptanone (0.816 μg/mL internal standard) were placed in a headspace vial (20 mL, Beijing Banxia Technology Development Co., Ltd., Beijing, China). The stock was equilibrated in a constant temperature water bath at 60 °C for 20 min, and then the volatile compounds were extracted with SPME needles having divinylbenzene/carboxyl/polydimethylsiloxane fibers (50/30 μm, Supelco, Bellefonte, PA, USA) at 60 °C for 40 min. Immediately after the extraction was completed, the extraction needle was inserted into the GC-O-MS and GC × GC-O-MS instruments for analysis and thermal desorption at 250 °C for 5 min. For accuracy, each sample was analyzed three times.

### 2.3. Solvent-Assisted Flavor Evaporation (SAFE)

The SAFE system is a compact combination of a distillation unit and a high vacuum pump; 40 g of fish soup and 5 μL of 2-methyl-3-heptanone (0.816 μg/μL, internal standard) was added to a Teflon bottle with diethyl ether/pentane (V1/V2 = 2:1). The resulting preparation was extracted by stirring for 8 h on a shaker (4 °C), and the volatile odor compounds were extracted using a SAFE apparatus (Deutschen Forschungsanstalt für Lebensmittelchemie, Freising, Free State of Bavaria, Germany). The distillation process was carried out by a molecular turbine pump (Edwards, Munich, Germany) for 2 h at 10-4 torr. The resulting fractions were collected in a trap that was submerged under liquid nitrogen. After collection, water was removed from the collected fraction by adding anhydrous Na2SO4 and the collected fraction was concentrated to 10 mL through a Vigreux column (50 cm × 1 cm I.D.; Beijing Banxia Technology Development Co., Ltd., Beijing, China). The concentrate was further reduced to 200 μL by a nitrogen stream (purity ≥ 99.999%) purging. Finally, one microliter of the sample was injected into the GC-O-MS and GC × GC-O-MS instruments for detection using a 5 μL syringe. All the samples were analyzed in triplicates for statistical accuracy.

### 2.4. Dynamic Headspace Sampling (DHS)

Fish soup (20 g) and 2 μL of 2-methyl-3-heptanone (0.816 μg/μL, internal standard) were added to a dynamic headspace flask, which was purged with 99.9992% high purity nitrogen at a flow rate of 150 mL/min at one end and inserted into a Tenax TA tube at the other end for adsorption. The temperature of water bath was established at 60 °C, equilibrated for 20 min, and the adsorption was performed for 40 min. After that, the Tenax TA tube was removed and placed in a thermal desorption unit (TDU) (Gerstel, Germany) for GC-O-MS or GC × GC-O-MS analysis. For statistical accuracy, all the samples were analyzed in triplicates.

### 2.5. Stir Bar Sorptive Extraction (SBSE)

Fish soup (15 g) and 1.5 μL of 2-methyl-3-heptanone (0. 816 μg/μL, internal standard) were added to a 40 ml headspace vial, and a Twister® (Gerstel, Germany) (such as PDMS, EG (ethylene glycol) or silica gel) was immersed inside the headspace vial (carrying the sample) and stirred at 60 °C for 40 min. After that, the stirring rod was picked out, rinsed with deodorized water, and placed in a hollow glass tube, which was further transferred to a TDU for thermal desorption in order to perform GC-O-MS and GC × GC-O-MS analysis. For accuracy, all the samples were analyzed three times.

### 2.6. Liquid-Liquid Extraction (LLE)

The volatile substances in the samples were extracted according to the conventional liquid-liquid extraction (LLE) method [15]. Fifty grams of sample was transferred to a triangular flask, after that, 50 mL distilled water, 50 mL dichloromethane, 50 mL diethyl ether and 5 μL of internal standard 2-methylhept-3-one (0. 816 μg/μL, internal standard) were added together. The mixture was stirred at 800 rpm for 10 min. After centrifugation at 8,000g for 40 minutes, the extracts were separated using a partition funnel. The resulting extract was added with anhydrous sodium sulfate and left to dry at 4 °C for 12 h. The dried extract was then concentrated to 100 μL using a nitrogen stream, and finally the concentrated extract was analyzed by GC-O-MS and GC × GC-O-MS. The samples were analyzed in triplicates for accuracy and statistical analysis. 

### 2.7. Gas Chromatography-Olfactometry-Mass Spectrometry (GC-O-MS) Analysis 

A GC-MS (7890A-7000, Agilent Technologies Inc., Santa Clara, CA, USA) instrument combined with an olfactory detection port (ODP4, Gerstel, Germany) was used to identify volatile odor compounds. Separation of odor-active substances in samples was performed on a polar DB-WAX capillary column (30 mm × 0.32 mm, 0.25 μm film thickness; J & W Scientific, Folsom, CA, USA). The gas chromatographic instrument condition includes an initial column temperature setting of 40 °C, holding for 3 min, followed by an increase in temperature up to 230 °C at 4 °C/min and holding for 3 min. Ultra-pure helium (99.999%, Beijing AP BAIF Gas Industry Co., Ltd., Beijing, China) was used as the carrier gas. The electron impact mass spectra were generated at an ionization energy of 70 eV with an *m*/*z* scan range of 25–370 amu. The temperatures of the mass spectrometer source and quadrupole were programed at 230 °C and 150 °C, respectively. Moisture gas was delivered to the olfactory detection port through a blank capillary column.

### 2.8. GC × GC-O-MS Analysis

A GC-MS instrument (8890A-5977B; Agilent Technologies Inc., Santa Clara, CA, USA) with an olfactory detector (OFD) Sniffer 9100 (Brechbühler, Schlieren, Switzerland) was used to identify volatile odor compounds in the samples. Two columns were used to separate the components, DB-WAX (polar, 30 m × 0.25 mm × 0.25 µm; Agilent Technologies) and DB-17 (mid polar, 2.22 m × 0.18 mm × 0.18 µm; Agilent Technologies). The initial column temperature was established at 40 °C and held for 3 min, followed by a ramp-up to 230 °C at 4 °C/min, held at 230 °C for 3 min. Ultra-pure helium (99.999%, Beijing APB Gas Industry Co., Ltd., Beijing, China) was used as the carrier gas. The electron impact mass spectra were generated at an ionization energy of 70 eV with an *m*/*z* scan range of 35–375 amu. The temperatures of the mass spectrometer source and quadrupole were programmed at 230 °C and 150 °C, respectively. A solid-state modulator SSM1800 (J&X Technologies, Shanghai, China) was placed between the two columns for the heating and cooling phases. The temperature of the cold zone was kept at −50 °C and the modulation cycle was set to 5 s.

The concentrated fractions in the instrument were analyzed at the sniffing port by three trained sensory panelists who were members of the Molecular Sensory Laboratory from Beijing University of Technology; they were trained for four weeks to analyze the effluent components at the sniffing port for 2 h per day. During the GC-O analysis, moisture was delivered to the sniffing port through a blank capillary column. The odor descriptors and label odor intensity values, as well as retention times, were recorded [16]. The compounds that were perceived by two or more panelists were tentatively identified as odor compounds.

### 2.9. Aroma Extraction Dilution Analysis (AEDA)

AEDA was analyzed by varying the fractionation ratio of GC-O. The importance of each volatile component was ranked by the average flavor dilution (FD) factor, which was determined at the sniffing port according to the AEDA procedure. According to the ratios of 1:3n, which were 0, 1:3, 1:9, 1:27, 1:81, etc. The corresponding flavor dilution (FD) factors were defined as 1, 3, 9, 27, 81, etc. The result of the FD factor analysis refers to the maximum value for which the compound can be detected. Consequently, the FD factor of the odor compound that was analyzed at the sniffing port was 81, which indicates that the odor-active substance was not analyzed by the panelist at the sniffing port when the GC-O dilution ratio was 81:1. Usually, the higher the FD factor obtained for the odor-active compound, the more critical it is to be analyzed.

### 2.10. Qualitative and Quantitative Analysis

Volatile odor compounds were identified by mass spectrometry (MS) and comparison of identified peaks with the NIST library mounted on GC-MS. The linear retention index (RI) was calculated for dual identification according to the following equation:RI = 100N + 100n{[t Ra − t RN] / [t R( N + n) − t RN]}(1)
where N represented the number of carbon atoms with peaks in front of the identified compound, n represented the numerical difference between the upper and lower n-alkanes in gas chromatography, and the variables t Ra, t RN, t R (N + n) denoted the retention time of the identified compounds and the upper alkane and lower alkanes, respectively. The unknown odor compounds were positively identified by the three methods: (1) comparison of RI and odor descriptions (O) with reference compounds; (2) comparison of MS data with NIST library; (3) finally by standard (STD) compounds verification [17].

### 2.11. Relative Odor Activity Value (R-OAV) Calculation 

By adapting the method of Yang et al, the r-OAV was calculated using the following formula:r-OAV = Ci/OTi(2)
where Ci represented the relative concentration of a certain volatile odor compound in the fish soup and OTi denoted the odor threshold of the compound [18]. The threshold of the compound used in this study was the actual threshold in water [19]. 

### 2.12. Omission Experiment

An aroma model composed of the selected key odor-active compounds was prepared by reducing one of the 11 key aroma compounds at a time. A triangulation experiment assessed by a sensory panel was used to compare the differences between the blended quintile and the complete reconstituted fraction according to the PRC national standard. In other words, if 8, 9, and 10 or more of the 12 sensory members were able to identify differences between a compound when it was omitted and completed the reconstitution model, the results were considered significant (α ≤ 0.05), highly significant (α ≤ 0.01), or very highly significant (α ≤ 0.001), respectively. If fewer than 8 were identified, the result was considered “not significant”. Sensory evaluation is a common method used in the food science area, which also does not involve informed consent and ethics.

### 2.13. Statistical Analysis

All experiments were performed in triplicate, and the data was expressed as mean ± standard deviation. Statistical analysis was performed using Microsoft Excel 2019 (Microsoft Corp., Redmond, WA, USA).

## 3. Results and discussion

### 3.1. Comparison of Different Extraction Methods for Analysis of Odor Compounds

Based on the efficiencies of the five extraction methods including SPME, SAFE, DHS, SBSE and LLE the best extraction method was selected and the odor substances in the fish soup were analyzed by GC-O-MS and GC × GC-O-MS. As shown in Table 1, in the 1D GC mode, a total number of 38 odor compounds were extracted by SPME, including 5 alcohols, 15 aldehydes, 7 ketones, 6 acids and 5 others. Ten odor compounds were detected from SAFE extract, including 3 alcohols, 2 aldehydes, 1 ketone, 1 acid and 3 others. Eighteen different types of odor compounds were extracted by DHS, including 8 aldehydes, 3 ketones, 2 esters, 2 acids and 3 others. Whereas, 11 odor compounds were extracted from SBSE, including 6 aldehydes, 2 ketones, 2 esters and 1 acid. While for LLE, only one aldehyde and one ketone were extracted. In the 2D GC mode. A total of 39 odor compounds were extracted by SPME, including 3 alcohols, 12 aldehydes, 9 ketones, 1 ester, 2 acids and 12 others; 33 odor compounds were extracted by SAFE, including alcohols (5), aldehydes (7), ketones (8), ester (1), acids (2) and others (10); by DHS, 21 volatiles were extracted including alcohols (2), aldehydes (8), ketones (4), esters (2), acids (3) and others (2). A total of 14 odor compounds were extracted from SBSE, including alcohol (1), aldehydes (4), ketones (2), ester (1), acids (3), and others (2); and only 10 volatile compounds were extracted through LLE, including alcohol (1), aldehydes (4), ketones (2) and esters (3). The results of the five extraction methods using 1D GC mode were compared, and the results are presented in Figure 1 and Figure 2; it can be clearly concluded that SPME extracted a higher number of odor compounds. 

To better determine the best method for the extraction of volatile compounds, OPLS-DA was chosen for further differentiation. As shown in Figure 3a,b, the OPLS-DA analysis showed that SPME was better differentiated from the other four extraction methods. In 1D mode, LLE could not be distinguished from SBSE. In 2D mode, three extraction methods cannot be distinguished, namely DHS, LLE and SBSE.

It has been shown that among the alcohols, 1-octen-3-ol is one of the potent compounds that contributed the most to the fishy odor [10,20], and it was predicted that lipoxygenases cause the breakdown of polyunsaturated fatty acids or the reduction of carbonyl compounds [21]. 1-Octen-3-ol was detected in SPME, SAFE and DHS, and the highest level (7.07 ± 0.98 μg/Kg) was detected when analyzed in the 2D mode in SPME; this was in agreement with the study of Xue et al. [10]. However, the aforementioned compound was not detected in both the SBSE and LLE extraction methods. Aldehydes are a relatively large group of compounds that contribute to fish odor [22]. Some of the aldehydes that can be easily sniffed were detected in SPME, but almost no aldehydes were extracted via LLE; this also confirms that LLE is not suitable for the analysis of odor compounds in fish. The extraction of odorous substances from fish by the LLE method has not been reported in recent studies. 1-Octen-3-one has the same mushroom flavor as 1-octen-3-ol, which has a low threshold and can be distinctly sniffed; it is worth noting that this substance is only detected in the 2D mode of SPME and is not detected by any other method. A few previous reports have shown 1-octen-3-one as a fishy substance, but to the best knowledge of the authors, this is the first time that 1-octen-3-one has been identified in fish. Pyridine was once considered as one of the main fishy substances in freshwater fish [23]. However, it has not been reported in recent studies [24,25,26]; this may be related to the detection method. In this study, only the 2D mode of SAFE was detectable at minimal levels and was not detected by olfaction. In addition, small amounts of pyrazines and furans were detected in SPME, DHS and SAFE, probably due to substances generated by prolonged high temperature heating during the boiling process [27], and the contents in SPME were higher than the other two methods. In summary, the best choice was concluded as SPME.

### 3.2. Comparison of Aroma Compounds between GC-O-MS and GC × GC-O-MS Analysis

For a clearer comparison of the methods of extracting volatile compounds and analysis of odor-active substances in fish soup, GC-O-MS (1D, one dimension) and GC × GC-O-MS (2D, two dimension) were employed. For this, the results of the two-dimensional mode (GC × GC-O-MS) were compared with the analysis of all volatile compounds detected through the one-dimensional column (first column). Compared with GC-O-MS, GC × GC-O-MS can analyze the odor compounds in the sample more accurately based on its high resolution and high sensitivity. In other words, the GC × GC-O-MS analysis can be used to confirm whether each odor region is shared by multiple odor compounds; and it can also be used to quickly distinguish the compounds in multiple odor regions. Therefore, the GC × GC-O-MS method can be used to analyze volatile odor compounds more efficiently than GC-O-MS, although the analytical results of both instruments show a high degree of agreement.

As shown in Figure 4a–e, among the five extraction methods, more compounds were detected in 2D mode than in 1D mode. As shown in Table 1, the relative content of the compounds analyzed by the 2D model was also higher. The 3D view of the 2D mode was presented in Figure 5a,b. Specifically, 9 compounds, such as decanal, dimethyl trisulfide, and 2,5-dimethylpyrazine, were not analyzed in the 1D mode, but several of these substances could be identified in the 2D mode and could be analyzed in the sniffing port. Interestingly, it was observed that 1-octen-3-one can be smelled but not identified in the 1D mode, which does not reach the detection limit of the instrument, while it can be both analyzed and smelled in the 2D mode, which further confirms the high resolution of the 2D mode. In addition, other substances, such as 1-octen-3-ol, acetic acid, and (E)-2-octenal, can be analyzed in the 1D mode but cannot be identified in the sniffing port, while these compounds can be easily analyzed and sniffed in the 2D mode. Some compounds such as furfural, (E,Z)-2,6-nonadienal, and 4-hydroxy-2,5-dimethyl-3(2H) furanone, which were not identified in 2D mode, could be analyzed or smelled in the 1D mode. Clearly, the above results revealed that the use of both analytical methods can analyze the odor compounds in the samples more comprehensively.

### 3.3. Key Aroma-Active Compounds Identified by AEDA and r-OAV

It was found that all the odor compounds in the sample contributed to the overall odor profile. To analyze the compounds that contributed more, a dilution analysis was performed in order to identify the key odor compounds in the sample. The critical level of the odor-active compounds was determined by the FD factor obtained from the dilution analysis. In other words, the FD factor is positively correlated with the contribution of the odor compounds to the overall odor profile [28].

By using SPME combined with GC-O-MS and GC × GC-O-MS, 55 odor compounds were identified, including 5 alcohols, 19 aldehydes, 12 ketones, 1 ester, 6 acids, 5 pyrazines, 3 furans, and 6 others (Table 1), and their FD factors were obtained by AEDA, as shown in Table 2. Among these, 2-pentylfuran, and 1-octen-3-one showed the highest FD factor (FD = 729), followed by 4-hydroxy-2,5-dimethylfuran-3(2H)-one, (E)-2-octenal, 1-octen-3-ol and hexanal with FD factors of 243. However, hexanal, 2-pentylfuran, 1-octen-3-one, (E)-2-octenal, and 1-octen-3-ol contributed significantly to the odor characteristics of the fish soup; these results were consistent with previous studies [10]; these compounds delivered the odor of mushroom, beany, grass and cucumber, creating a characteristic odor in fish soup. The FD factors of these odor-active compounds were then obtained, including 6-methylhept-5-en-2-one (81), (E,Z)-2,6-nonadienal (27), 2-ethyl-3,5-dimethylpyrazine (27) and 3-methylpentanoic acid (27). Since these compounds showed a high FD factor, they were considered to be the key odorants in fish soup. However, some typical substances that usually affect the fish odor, such as heptanal, nonanal and trimethylamine, did not contribute to the overall odor profile and were not even detected in this study, probably due to their changes in content after boiling; this was for the first time that 3-methylpentanoic acid was detected as a compound with sweaty odor with an FD factor of 9, and also as a contributing compound to the fish odor. Importantly, the advantages of GC × GC-O-MS in the detection of odor compounds were also confirmed. In addition, the FD factor for acetic acid, 2,6-dimethylpyrazine and 5-methyl-2-thiophenecarboxaldehyde was 9, while furfural and dimethyl trisulfide had an FD factor of 3. Although the FD factors of these compounds are relatively small compared to other substances, since they were also smelled at the sniffing port, so it was predicted that they also made contributions to the overall odor profile of the fish soup.

The concentrations of the compounds after AEDA analysis were calculated by a semi-quantitative method. Table 2 summarizes the concentrations as well as the results of r-OAV calculations. Among the odor compounds, 1-octen-3-one had the highest r-OAV of 7500, attributed to the fishy odor; this result was also consistent with the previous results [29]. Secondly, 2-pentylfuran had a higher r-OAV of 3436, and its odor attribute was beany; this compound has not been previously reported in the literature as a fishy odorant and may be generated from fatty acids during the boiling of fish scraps. (E)-2-Octenal, hexanal, 4-hydroxy-2,5-dimethylfuran-3(2H)-one, 1-octen-3-ol, and 6-methylhept-5-en-2-one had the r-OAVs of 1452, 1212, 904, 869 and 602, respectively. The r-OAVs of these compounds were consistent with the AEDA results. Additionally, these compounds were previously reported as fishy substances [30]. However, there was one exception, such as 6-methylhept-5-en-2-one with an FD factor of 81, but it showed a lower FD factor relative to other substances with an FD factor of 27 (2-ethyl-3,5-dimethylpyrazine and (E,Z)-2,6-nonadienal). The results of the above analysis indirectly suggested the existence of some synergistic effect between odor compounds, rather than a single relationship, which could potentially enhance or diminish the overall aroma; these results better illustrate the importance of olfactory detection when analyzing the odor compounds.

### 3.4. Omission Experiment

Aroma model was constructed by selecting the top 11 key odor compounds in order of r-OAV value. Triangle tests were performed for the deletion of the 11 key odor compounds one by one and the results are presented in Table 3. The absence of 2-pentylfuran, (E)-2-octenal, 1-octen-3-one and hexanal was detected by all panelists, which indicates the importance of these compounds in contributing to the overall odor profile. While in the absence of acetic acid, 4-hydroxy-2,5-dimethylfuran-3(2H)-one, and 3-methylpentanoic acid, the results showed insignificant, i.e., less impact on the overall aroma. Thus, the omission experiments further confirmed that 2-pentylfuran, (E)-2-octenal, 1-octen-3-one, hexanal, 1-octen-3-ol, 6-methylhept-5-en-2-one, (E,Z)-2,6-nonadienal and 2-ethyl-3,5-dimethylpyrazine were the key odor compounds in the fish soup.

## 4. Conclusions

In this study, fish scraps were used as raw materials for the preparation of fish soup. Five volatile extraction methods were compared in order to select the best method for extracting odor compounds from fish soup. Finally, SPME combined with GC-O-MS and GC × GC-O-MS were selected to analyze odor-active compounds together. A total of 57 volatile compounds were identified. Among these, 38 volatile compounds were detected by GC-O-MS, of which 10 were declared as odor-active compounds, i.e., substances that could be smelled at the sniffing port; while 39 volatile compounds were detected by GC × GC-O-MS, of which 12 were odor-active. The above results revealed that the combination of these two analytical methods can provide a more comprehensive analysis of the volatile aroma compounds in fish soup. The results of AEDA analysis showed that 1-octen-3-one and 2-pentylfuran contributed most significantly to the odor of fish soup, followed by (E)-2-octenal, hexanal, 4-hydroxy-2,5-dimethylfuran-3(2H)-one, 1-octen-3-ol, 2-ethyl-3,5-dimethylpyrazine, (2E,6Z)-nona-2,6-dienal, 6-methylhept-5-en-2-one and 3-methylvaleric acid. From the r-OAV results, it was concluded that the compound with the highest r-OAV was 1-octen-3-one, followed by 2-pentylfuran. In summary, 2-pentylfuran, (E)-2-octenal, 1-octen-3-one, hexanal, 1-octen-3-ol, 6-methylhept-5-en-2-one, (E,Z)-2,6-nonadienal and 2-ethyl-3,5-dimethylpyrazine were the most important odor-active compounds in the fish soup as confirmed by omission experiment and triangle test. In the subsequent studies, based on the fishy compounds in this study, enzymatic digestion of fish soup using protease was performed to remove the fishy compounds and enhance the umami taste of the fish soup, contributing to the production of a full-flavored and distinctive seasoning; this approach not only improves the comprehensive utilization value of fish scraps, but also avoids the environmental pollution caused by discarding fish scraps.

## Figures and Tables

**Figure 1 molecules-27-06177-f001:**
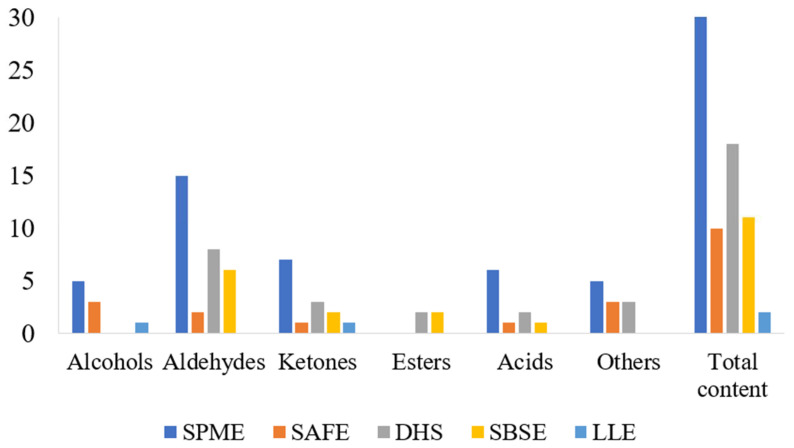
The number of volatile compounds measured by GC-O-MS for five extraction methods.

**Figure 2 molecules-27-06177-f002:**
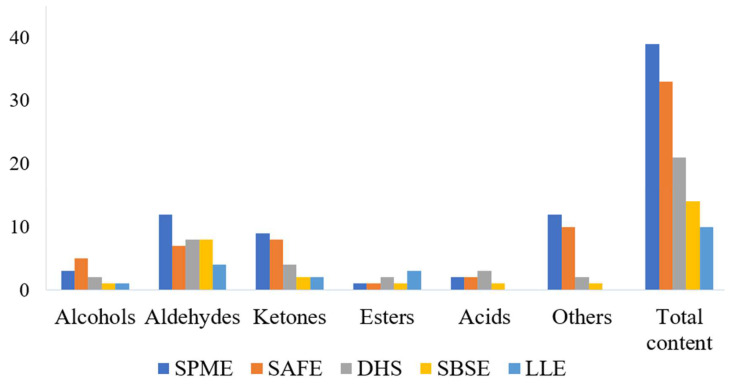
Number of volatile compounds measured by GC × GC-O-MS for five extraction methods.

**Figure 3 molecules-27-06177-f003:**
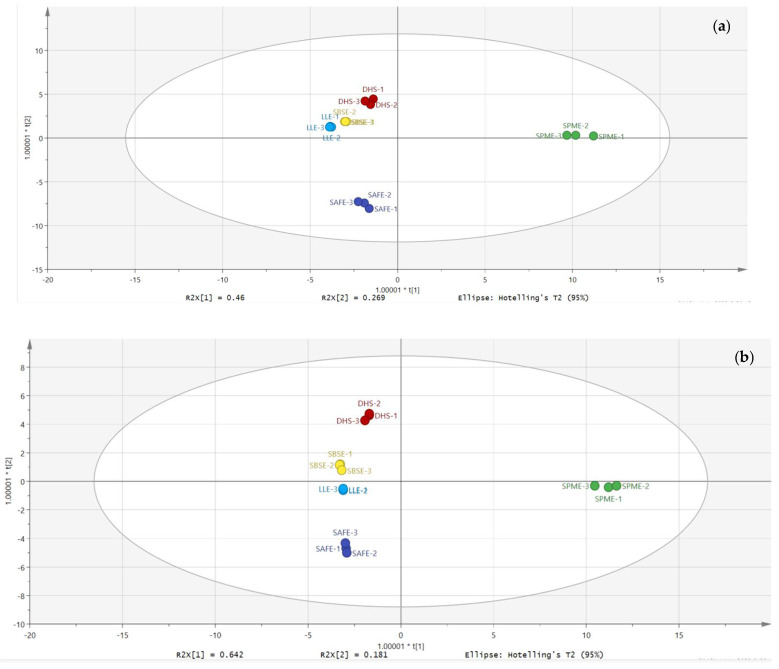
OPLS-DA analysis chart of five extraction methods. (**a**) OPLS-DA analysis of five extraction methods in 1D mode. (**b**) OPLS-DA analysis of five extraction methods in 2D mode.

**Figure 4 molecules-27-06177-f004:**
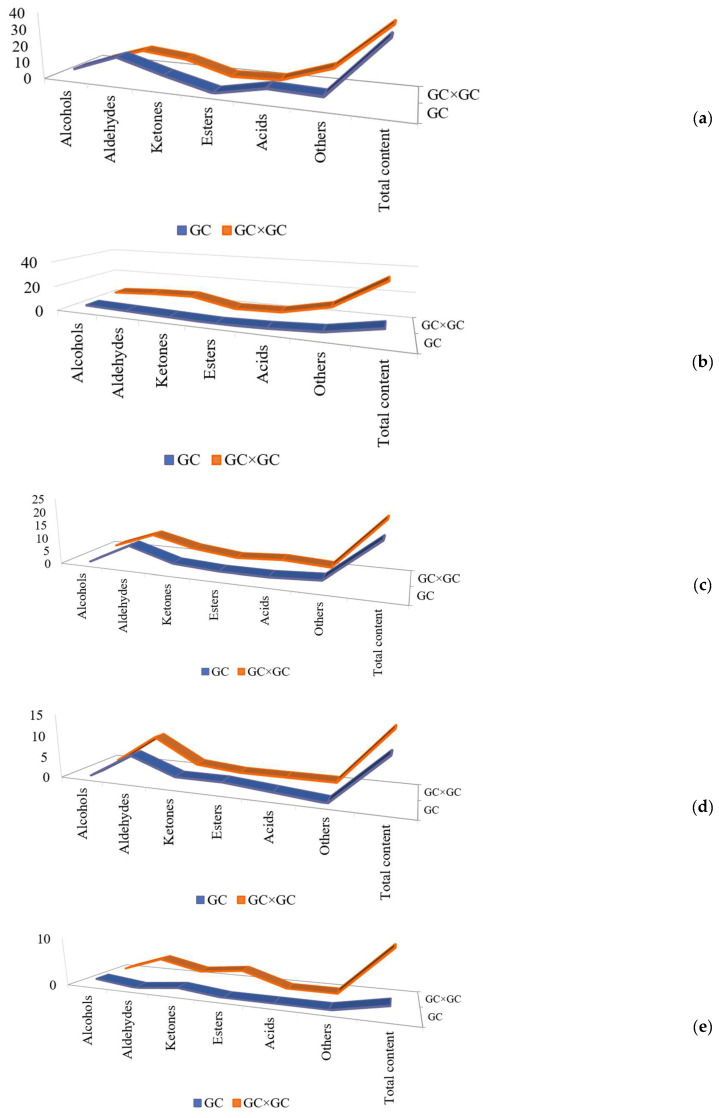
Five extraction methods combined with GC and GC × GC to measure the amount of volatile compounds. (**a**) SPME combined with GC and GC × GC to measure the amount of volatile compounds; (**b**) SAFE combined with GC and GC × GC to measure the amount of volatile compounds; (**c**) DHS combined with GC and GC × GC to measure the amount of volatile compounds; (**d**) SBSE combined with GC and GC × GC to measure the amount of volatile compounds; (**e**) LLE combined with GC and GC × GC to measure the amount of volatile compounds.

**Figure 5 molecules-27-06177-f005:**
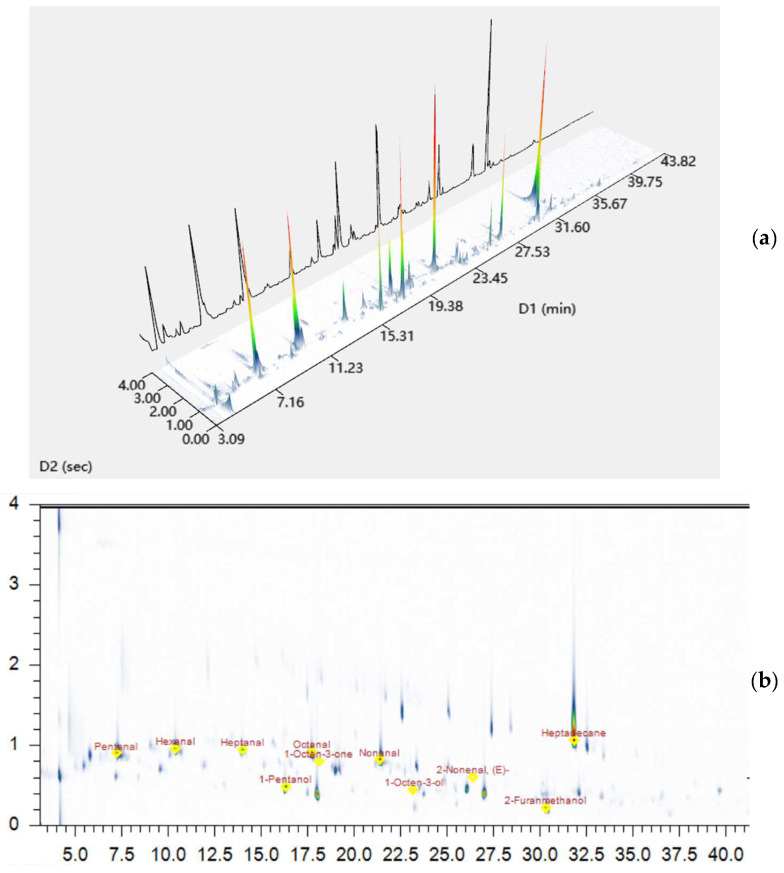
(**a**) The 3D analytical images of GC × GC-O-MS in sample; (**b**) The 2D analytical images of GC × GC-O-MS in sample.

**Table 1 molecules-27-06177-t001:** Volatile compounds identified by five extraction methods in combination with GC-O-MS and GC×GC-O-MS.

No.	Component	CAS	Odor	RI ^a^		Method of Identification ^b^		Relative Content (ng/g) ^c^									
								SPME		SAFE		DHS		SBSE		LLE	
				GC	GC × GC	GC	GC × GC	GC	GC × GC	GC	GC × GC	GC	GC × GC	GC	GC × GC	GC	GC × GC
Alcohols																	
1	1-Pentanol	71-41-0	balsam	1247	1239	MS/RI	MS/RI	50.54 ± 4.37	53.99 ± 6.18	ND	41.27 ± 12.73	ND	ND	ND	ND	ND	ND
2	2-Furanmethanol	98-00-0	musty	1702	1697	MS/RI	MS/RI	229.61 ± 16.49	288.19 ± 23.04	165.24 ± 26.33	ND	ND	203.15 ± 35.86	ND	ND	ND	75.09 ± 8.39
3	2-Thiophenemethanol	636-72-6	roasted	1885	ND	MS/RI	ND	23.83 ± 1.83	ND	ND	ND	ND	ND	ND	ND	ND	ND
4	1-Octen-3-ol	3391-86-4	mushroom	1430	1418	MS/RI	MS/RI/O	6.08 ± 1.08	7.07 ± 0.98	ND	5.49 ± 2.04	ND	6.57 ± 4.19	ND	ND	ND	ND
5	1-Butanol	71-36-3	oil	ND	1126	ND	MS/RI	ND	ND	ND	ND	ND	ND	ND	ND	ND	ND
6	Linalool	78-70-6	floral	ND	1509	ND	MS/RI	ND	ND	ND	12.48 ± 1.06	ND	ND	ND	ND	ND	ND
7	α-Cumyl alcohol	617-94-7	green	1743	ND	MS/RI	ND	ND	ND	2.05 ± 0.16	ND	ND	ND	ND	ND	0.16 ± 0.11	ND
8	2-Ethyl-1-hexanol	104-76-7	citrus	ND	1492	ND	MS/RI	ND	ND	ND	ND	ND	ND	ND	0.96 ± 0.06	ND	ND
Aldehydes																	
9	Furfural	98-01-1	sweet woody	1493	ND	MS/RI/O	ND	117.6 1± 5.40	ND	ND	ND	ND	ND	ND	ND	ND	ND
10	5-Methyl-2-thiophenecarboxaldehyde	13679-70-4	almond	1764	ND	MS/RI/O	ND	33.50 ± 1.57	ND	ND	ND	ND	ND	ND	ND	ND	ND
11	(*E*,*Z*)-2,6-Nonadienal	557-48-2	green fatty	1590	ND	MS/RI/O	ND	3.06 ± 0.05	ND	ND	ND	ND	ND	ND	ND	ND	ND
12	2-Methylbutyraldehyde	96-17-3	musty cocoa	987	ND	MS/RI	ND	25.10 ± 3.72	ND	ND	ND	ND	ND	ND	ND	ND	ND
13	3-Methylbutanal	590-86-3	chocolate	924	935	MS/RI	MS/RI	110.01 ± 7.46	134.51 ± 11.36	ND	ND	58.19 ± 2.63	108.32 ± 4.27	ND	ND	ND	ND
14	Pentanal	110-62-3	fermented	956	944	MS/RI	MS/RI	159.20 ± 4.02	201.26 ± 18.43	ND	ND	ND	ND	ND	ND	ND	ND
15	Hexanal	66-25-1	grass	1062	1058	MS/RI/O	MS/RI/O	92.66 ± 1.17	118.80 ± 7.34	ND	88.63 ± 9.33	66.04 ± 8.39	74.68 ± 6.62	26.59 ± 1.95	29.27 ± 2.29	ND	32.64 ± 5.15
16	Octanal	124-13-0	waxy citrus	1279	1265	MS/RI	MS/RI	2.49 ± 0.36	23.31 ± 1.26	ND	3.91 ± 0.73	ND	ND	0.27±0.06	3.08 ± 0.74	ND	ND
17	Nonanal	124-19-6	orange	1384	1391	MS/RI	MS/RI	235.93 ± 12.73	304.04 ± 25.62	175.56 ± 14.76	209.37 ± 12.94	104.27 ± 9.26	ND	ND	52.60 ± 2.46	ND	17.40 ± 0.95
18	(*E*)-2-Octenal	2548-87-0	cucumber	1437	1426	MS/RI	MS/RI/O	8.71 ± 0.76	9.56 ± 0.38	ND	ND	ND	ND	ND	ND	ND	ND
19	Benzaldehyde	100-52-7	sweet	1503	1509	MS/RI	MS/RI	103.81 ± 8.25	144.38 ± 12.34	ND	73.29 ± 8.37	68.53 ± 10.55	71.27 ± 9.27	34.17 ± 2.91	63.26 ± 5.72	ND	9.63 ± 0.94
20	5-Methyl-2-furancarboxaldehyde	620-02-0	spice caramel	1578	1554	MS/RI	MS/RI	352.37 ± 18.45	329.13 ± 18.23	ND	164.24 ± 24.53	228.72 ± 13.66	303.24 ± 36.81	73.02 ± 10.95	142.88 ± 11.91	ND	ND
21	Phenylacetaldehyde	122-78-1	green	1662	ND	MS/RI	ND	13.68 ± 2.33	ND	ND	ND	ND	ND	ND	ND	ND	ND
22	Pentadecanal	2765-11-9	fresh waxy	2019	ND	MS/RI	ND	10.82 ± 0.12	ND	ND	ND	ND	ND	ND	ND	ND	ND
23	(*Z*)-4-Heptenal	6728-31-0	oily fatty	ND	1287	ND	MS/RI	ND	8.46 ± 1.04	ND	ND	ND	ND	ND	ND	ND	ND
24	Methional	3268-49-3	musty	ND	1439	ND	MS/RI	ND	41.43 ± 6.12	ND	ND	ND	ND	ND	ND	ND	ND
25	Decanal	112-31-2	sweet waxy	ND	1538	ND	MS/RI/O	ND	25.18 ± 2.25	ND	9.73 ± 0.92	ND	11.04 ± 1.52	3.16 ± 0.06	3.95 ± 0.31	ND	ND
26	5-Methyl-2-thiophenecarboxaldehyde	13679-70-4	woody	ND	1783	ND	MS/RI/O	ND	20.34 ± 1.59	ND	ND	ND	ND	ND	ND	ND	ND
27	Dodecyl aldehyde	112-54-9	soapy waxy	1745	ND	MS/RI	ND	ND	ND	18.77 ± 0.35	ND	ND	ND	ND	ND	ND	ND
28	(*E*,*E*)-2,4-Heptadienal	4313-03-5	green	ND	1497	ND	MS/RI	ND	ND	ND	6.38 ± 0.74	ND	ND	ND	ND	ND	ND
29	Heptaldehyde	111-71-7	fresh	1174	1165	MS/RI	MS/RI	ND	221.07 ± 14.28	ND	ND	84.93 ± 5.27	164.29 ± 12.63	43.28 ± 13.75	53.22 ± 2.76	ND	64.01 ± 6.37
30	3-Methylhexanal	19269-28-4	sweet green	1102	1116	MS/RI	MS/RI	ND	ND	ND	ND	5.37 ± 1.10	12.84 ± 2.12	ND	ND	ND	ND
31	2-Undecenal	2463-77-6	fruity	1793	1794	MS/RI	MS/RI	ND	ND	ND	ND	ND	ND	16.38 ± 0.97	9.96 ± 0.72	ND	ND
Ketones																	
32	Hydroxyacetone	116-09-6	caramel	1272	ND	MS/RI	ND	522.52 ± 48.29	ND	ND	ND	ND	ND	ND	ND	ND	ND
33	4-Hydroxy-2,5-dimethyl-3(2*H*)furanone	3658-77-3	sweet	2037	ND	MS/RI/O	ND	35.25 ± 1.45	ND	ND	ND	ND	ND	ND	ND	ND	ND
34	6-Methyl-5-hepten-2-one	110-93-0	citrus	1342	1332	MS/RI/O	MS/RI/O	4.34 ± 0.03	8.06 ± 1.24	ND	2.96 ± 0.91	7.47 ± 0.26	9.26 ± 0.27	0.38 ± 0.05	5.67 ± 0.11	ND	ND
35	2-Butanone	78-93-3	camphor	901	914	MS/RI	MS/RI	22.05 ± 9.25	29.37 ± 1.96	ND	19.62 ± 2.28	ND	ND	ND	ND	ND	ND
36	2,3-Pentanedione	600-14-6	pungent	1047	1058	MS/RI	MS/RI	7.17 ± 0.45	16.54 ± 0.52	ND	ND	ND	ND	ND	ND	ND	ND
37	Acetoin	513-86-0	sweet	1284	1273	MS/RI	MS/RI	31.42 ± 3.72	35.21 ± 0.74	ND	17.82 ± 3.32	27.37 ± 3.56	29.52 ± 2.17	ND	ND	ND	5.83 ± 0.64
38	2-Dodecanone	6175-49-1	fruity	1669	ND	MS/RI	ND	0.42 ± 0.07	ND	ND	ND	ND	ND	ND	ND	ND	ND
39	5-Ethyl-2(5*H*)-furanone	2407-43-4	spice	1756	ND	MS/RI	ND	5.69 ± 1.65	ND	ND	ND	ND	ND	ND	ND	ND	ND
40	2,3-Butanedione	431-03-8	butter	ND	961	ND	MS/RI	ND	46.93 ± 4.38	ND	ND	ND	ND	ND	ND	ND	ND
41	1-Penten-3-one	1629-58-9	peppery	ND	1018	ND	MS/RI	ND	6.14 ± 0.47	ND	ND	ND	ND	ND	ND	ND	ND
42	2-Heptanone	110-43-0	fruity	ND	1165	ND	MS/RI	ND	2.17 ± 0.02	ND	ND	ND	ND	ND	ND	ND	ND
43	1-Hydroxy-2-propanone	116-09-6	pungent	1310	1307	MS/RI/O	MS/RI	ND	492.86 ± 42.74	ND	318.74 ± 41.17	148.02 ± 16.14	358.60 ± 22.61	ND	ND	74.26 ± 12.37	169.60 ± 19.95
44	1-Octen-3-one	4312-99-6	mushroom	ND	1208	O	MS/RI/O	ND	0.12 ± 0.01	ND	ND	ND	ND	ND	ND	ND	ND
45	Acetophenone	98-86-2	sweet	1643	1637	MS/RI	MS/RI	ND	ND	0.97 ± 0.13	3.02 ± 1.05	ND	ND	ND	ND	ND	ND
46	2-Decanone	693-54-9	orange	ND	1498	ND	MS/RI	ND	ND	ND	1.34 ± 0.07	ND	ND	ND	ND	ND	ND
47	1-Hydroxy-2-butanone	5077-67-8	coffee	ND	1397	ND	MS/RI	ND	ND	ND	0.94 ± 0.19	ND	ND	ND	ND	ND	ND
48	2-Tridecanone	593-08-8	fatty	ND	1809	ND	MS/RI	ND	ND	ND	9.48 ± 0.17	ND	ND	ND	ND	ND	ND
49	Geranylacetone	3796-70-1	green	ND	1853	ND	MS/RI	ND	ND	ND	ND	ND	7.47 ± 0.65	ND	ND	ND	ND
50	4-Hexen-3-one	2497-21-4	metallic	1187	1164	MS/RI/O	MS/RI/0	ND	ND	ND	ND	ND	ND	16.33 ± 3.61	15.27 ± 4.21	ND	ND
51	5-Ethyldihydro-2(3*H*)-furanone	695-06-7	herbal	1692	1687	MS/RI	MS/RI	ND	9.11 ± 1.03	ND	ND	ND	ND	2.09 ± 0.95	7.14 ± 1.27	ND	ND
Esters																	
52	Triethyl phosphate	78-40-0	cider	ND	1674	ND	MS/RI	ND	ND	ND	0.14 ± 0.03	ND	ND	ND	ND	ND	ND
53	Allyl 2-ethylbutyrate	7493-69-8	nut	1252	1246	MS/RI/O	MS/RI/O	ND	ND	ND	ND	9.27 ± 1.15	9.83 ± 1.63	ND	ND	ND	ND
54	geranyl isovalerate	109-20-6	green	1934	1948	MS/RI/O	MS/RI/O	ND	ND	ND	ND	14.38 ± 1.91	47.26 ± 3.25	21.22 ± 1.57	27.81 ± 4.38	ND	ND
55	Nitrous acid, ethyl ester	109-95-5	sweet	ND	1417	ND	MS/RI	ND	ND	ND	ND	ND	ND	ND	ND	ND	0.56 ± 0.27
56	Acetic acid, butyl ester	123-86-4	banana	ND	1049	ND	MS/RI	ND	ND	ND	ND	ND	ND	ND	ND	ND	19.47 ± 3.26
57	Butyrolactone	96-48-0	oily	ND	1634	ND	MS/RI	ND	ND	ND	ND	ND	ND	ND	ND	ND	5.22 ± 0.84
Acids																	
58	Propionic acid	79-09-4	pungent	1509	1528	MS/RI/O	MS/RI	20.387 ± 2.65	ND	ND	ND	ND	ND	ND	ND	ND	ND
59	3-Methylpentanoic Acid	105-43-1	sweaty	1761	ND	MS/RI/O	ND	49.50 ± 2.73	ND	ND	15.63 ± 3.68	ND	ND	ND	ND	ND	ND
60	Acetic acid	64-19-7	sour	1452	1433	MS/RI/O	MS/RI/O	487.86 ± 36.94	461.79 ± 54.27	ND	ND	268.38 ± 34.76	322.47 ± 29.81	173.28 ± 14.25	236.73 ± 25.77	ND	ND
61	Butyric acid	107-92-6	cheese	1631	1642	MS/RI	MS/RI	38.72 ± 6.28	41.76 ± 3.14	ND	ND	ND	ND	ND	ND	ND	ND
62	Hexanoic acid	142-62-1	fatty	1859	1846	MS/RI	MS/RI	69.13 ± 4.92	ND	ND	ND	27.48 ± 3.25	36.91 ± 2.44	ND	ND	ND	ND
63	Octanoic acid	124-07-2	waxy	2056	ND	MS/RI	ND	27.44 ± 1.74	ND	ND	ND	ND	ND	ND	ND	ND	ND
64	3-Methylbutanoic acid	503-74-2	sweaty	1646	ND	MS/RI/O	ND	ND	ND	ND	7.30 ± 1.93	ND	ND	ND	ND	ND	ND
65	Valeric acid	109-52-4	sickening	ND	1850	ND	MS/RI	ND	ND	ND	ND	ND	0.78 ± 0.16	ND	ND	ND	ND
Others																	
66	Styrene	100-42-5	sweet	1264	1239	MS/RI	MS/RI	9.00 ± 0.37	ND	5.46 ± 1.58	17.29 ± 2.73	ND	ND	ND	ND	ND	ND
67	2-Ethenyl-6-methylpyrazine	13925-09-2	hazelnut	1486	1502	MS/RI	MS/RI	21.00 ± 1.25	28.09 ± 3.38	ND	8.63 ± 1.35	ND	ND	ND	ND	ND	ND
68	2-Pentylfuran	3777-69-3	fruity	1231	1226	MS/RI/O	MS/RI/O	19.93 ± 0.05	19.46 ± 2.77	ND	ND	ND	ND	ND	ND	ND	ND
69	2,6-Di-tert-butyl-4-methylphenol	128-37-0	camphor	1905	ND	MS/RI	ND	6.33 ± 0.42	ND	ND	ND	ND	ND	ND	ND	ND	ND
70	2,6-Dimethylpyrazine	108-50-9	cocoa	1309	ND	MS/RI	ND	51.84 ± 10.47	ND	ND	ND	ND	ND	ND	ND	ND	ND
71	Pyrrole	109-97-7	sweet	ND	1528	ND	MS/RI	ND	11.89 ± 0.99	ND	ND	ND	ND	ND	ND	ND	ND
72	Dimethyl disulfide	624-92-0	vegetable	ND	1082	ND	MS/RI	ND	33.50 ± 1.16	ND	18.43 ± 2.55	ND	ND	ND	ND	ND	ND
73	Dimethyl trisulfide	3658-80-8	sulfurous	ND	1365	ND	MS/RI/O	ND	6.04 ± 0.53	ND	0.58 ± 0.17	ND	ND	ND	ND	ND	ND
74	Benzene	71-43-2	aromatic	ND	955	ND	MS/RI	ND	7.39 ± 0.56	ND	7.73 ± 0.36	ND	ND	ND	ND	ND	ND
75	2-Ethylfuran	3208-16-0	burnt	ND	945	ND	MS/RI	ND	11.25 ± 0.97	ND	ND	ND	ND	ND	ND	ND	ND
76	2,5-Dimethylpyrazine	123-32-0	roasted	ND	1329	ND	MS/RI/O	ND	63.21 ± 3.24	ND	ND	ND	ND	ND	ND	ND	ND
77	2,6-Dimethylpyrazine	108-50-9	cocoa	1352	1351	MS/RI	MS/RI	ND	1.36 ± 0.17	ND	0.06 ± 0.02	1.63 ± 0.16	5.37 ± 0.68	ND	ND	ND	ND
78	3-Ethyl-2,5-dimethylpyrazine	13360-65-1	potato	ND	1430	ND	MS/RI/O	ND	34.18 ± 1.25	ND	ND	ND	ND	ND	ND	ND	ND
80	Toluene	108-88-3	sweet	1039	1034	MS/RI	MS/RI	ND	ND	3.28 ± 0.53	7.58 ± 0.95	ND	ND	ND	ND	ND	ND
81	1,2-Xylene	95-47-6	geranium	1171	1164	MS/RI	ND	ND	ND	7.94 ± 1.27	ND	ND	ND	ND	ND	ND	ND
82	Pyridine	110-86-1	sour	ND	1169	ND	MS/RI	ND	ND	ND	0.93 ± 0.03	ND	ND	ND	ND	ND	ND
84	Naphthalene	91-20-3	pungent	ND	1701	ND	MS/RI	ND	ND	ND	16.39 ± 1.55	ND	ND	ND	ND	ND	ND
85	Benzothiazole	95-16-9	sulfury	1933	1930	MS/RI	MS/RI	ND	ND	ND	ND	0.27 ± 0.06	3.05 ± 0.74	ND	ND	ND	ND
86	Dimethyl trisulfide	624-92-0	onion	ND	1365	ND	MS/RI/O	ND	1.69 ± 0.03	ND	ND	ND	ND	ND	ND	ND	ND

a: RI, The Retention index on capillaries DB-WAX and DB-5. b: The different identification methods, including MS, RI, and O. In particular, for those compounds which were not able to identify by all three methods, they would be regarded as temporarily identified. ND, not detected. c: Relative concentration stated as the mean ± SD and the unit is ppt (part per trillion, ng/g).

**Table 2 molecules-27-06177-t002:** FD factors and relative odor activity values (r-OAV) of key aroma compounds.

No.	RI ^a^		Odor	Compound	FD Factors ^b^	Odor Threshold in Water (mg/kg) ^c^	r-OAV
	DB-WAX	DB-5					
1	1279	974	mushroom	1-Octen-3-one	729	0.000016	7500
2	1209	992	fruity	2-Pentylfuran	729	0.0058	3436
3	1440	1032	cucumber	(*E*)-2-Octenal	243	0.006	1452
4	1062	793	grass	Hexanal	243	0.098	1212
5	1960	1112	sweet	4-Hydroxy-2,5-dimethylfuran-3*(2H)*-one	243	0.039	904
6	1424	957	mushroom	1-Octen-3-ol	243	0.007	869
7	1430	1092	potato	3-Ethyl-2,5-dimethylpyrazine	27	0.0086	486
8	1582	1178	green fatty	*(E,Z)*-2,6-nonadienal	27	0.008	383
9	1325	954	citrus	6-Methylhept-5-en-2-one	81	0.025	202
10	1761	951	sweaty	3-Methylpentanoic acid	27	0.28	177
11	1428	668	sour	Acetic acid	9	22	40
12	1309	944	cocoa	2,6-Dimethylpyrazine	9	1.5	35
13	1493	802	sweet woody	Furfural	3	23	5
14	1764	1149	almond	5-Methyl-2-thiophenecarboxaldehyde	9	-	-
15	1365	972	sulfurous	Dimethyl trisulfide	3	-	-

a: RI, The Retention index on capillaries DB-WAX and DB-5. b: The FD factors of all aroma compounds in three samples were calculated by varying the split ratio of the GC inlet. c: Odor thresholds were referenced from a book named Odor thresholds compilations of odor threshold values in air, water and other media (second enlarged and revised edition).

**Table 3 molecules-27-06177-t003:** Omission experiment.

No.	Compounds ^a^	N ^b^	Significant ^c^
1	2-Pentylfuran	12	***
2	1-Octen-3-one	11	***
3	Acetic acid	6	-
4	4-Hydroxy-2,5-dimethylfuran-3*(2H)*-one	4	-
5	(E)-2-Octenal	10	***
6	1-Octen-3-ol	11	**
7	Hexanal	12	***
8	6-Methylhept-5-en-2-one	10	**
9	*(E,Z)*-2,6-Nonadienal	8	*
10	2-Ethyl-3,5-dimethylpyrazine	8	*
11	3-Methylpentanoic acid	6	-

a: The compounds which r-OAV in Table 3 showing a significant difference among OF, PC and AO. b: Number of correct judgments from 12 assessors by the triangle test. c: Significance: “*”, significant (α ≤ 0.05); “**”, highly significant (α ≤ 0.01); “***”, very highly significant (α ≤ 0.001); “-”, non-significant.

## Data Availability

The data will be available at request.

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
