# Peer review of "Comparison of Different Volatile Extraction Methods for the Identification of Fishy Off-Odor in Fish By-Products"

_molecules, 2022, doi:10.3390/molecules27196177_

Round 1
Reviewer 1 Report
The article by Zhang et al. compared extraction methods for different VOCs from fish soup using 1D and 2D analytical methods. The article can be published after responding to the following comments-
1. Section 2.8 DB-17 is not a non-polar column. Its 50% phenyl and considered as mid polar column.
2. It was observed by the authors that SPME was able to extract higher number of VOCs than other extraction methods. However, it must be mentioned here that SPME can offer biased sampling across volatilities. VOCs with higher volatilities are trapped far less efficiently than those with lower volatilities due its over reliance on partition coefficients.
3. The authors should produce a GCxGC chromatogram indicating different classes of compounds.
Author Response
Dear editor,
Thank you very much for giving us this opportunity to revise our manuscript. We are also highly obliged to the reviewer for his constructive comments on this manuscript. We have read every comment very carefully and modified the manuscript accordingly.
We have answered all the questions from the reviewer in detail and these are listed below point by point.
Reviewers’ Comments to Author:
Reviewer: 1
- Section 2.8 DB-17 is not a non-polar column. Its 50% phenyl and considered as mid polar column.
Answer: Thank you for your suggestion. The polarity of DB-17 has been changed to a mid polar column in line 187.
- It was observed by the authors that SPME was able to extract higher number of VOCs than other extraction methods. However, it must be mentioned here that SPME can offer biased sampling across volatilities. VOCs with higher volatilities are trapped far less efficiently than those with lower volatilities due its over reliance on partition coefficients.
Answer: Thank you for the valuable advice. We also thought about this concern you raised at the beginning of the experiment, and worried that the experiment could not achieve the purpose. However, the omission experiment result obtained in this study was not problematic in the end. Many researchers use SPME in the analysis of odor compounds because the technique is sensitive and the calibration is most straightforward. Wang et al. used SPME-GC-MS to analyze odor-active compounds in Yongchuan douchi and 49 compounds were sniffed and identified. Headspace (HS) analysis of Coffee Silverskin odor-active compounds using SPME-GC-MS by Angeloni et al. has been used to fully characterize the aroma of this sample.
The solid-phase coating (DVB/CAR/PDMS) selected for the SPME in this study has a strong affinity for the extracted organic components, which improves the sensitivity of the analysis. In choosing this method, we referred to a large amount of literature to optimize the various parameters of this method. For example, the effect of temperature equilibration on the extraction results was taken into account when preparing the samples, and after calibration it was finally determined that the samples were equilibrated in a constant temperature water bath at 60°C for 20 minutes, and then extracted with SPME at 60°C for 40 minutes. To ensure that the method conditions are optimal and the best for the sample analysis.
References:
Wang, S., Chang, Y., Liu, B., Chen, H., Sun, B., Zhang, N. Characterization of the Key Aroma-Active Compounds in Yongchuan Douchi (Fermented Soybean) by Application of the Sensomics Approach. Molecules. 2021, 26, 3048. doi: 10.3390/molecules26103048
Angeloni, S., Scortichini, S., Fiorini, D., Sagratini, G., Vittori, S., Neiens, S.D., Steinhaus, M., Zheljazkov, V.D., Maggi, F., Caprioli, G. Characterization of Odor-Active Compounds, Polyphenols, and Fatty Acids in Coffee Silverskin. Molecules. 2020, 25, 2993. doi: 10.3390/molecules25132993
- The authors should produce a GCxGC chromatogram indicating different classes of compounds.
Answer: Thank you for your suggestion. Modified at Table 1 and Figure 5. The types of compounds are indicated in Table 1. Two-dimensional chromatogram with labeled names of volatile compounds has been added in Figure 5.
Reviewer 2 Report
It looks like a great article, but the author is missing the main parts of the compound identification process. I would like to see the retention indices of each compound too.
Author Response
It looks like a great article, but the author is missing the main parts of the compound identification process. I would like to see the retention indices of each compound too.
Answer: Sincerely thanks for your valuable advice. I have made changes. The retention indices of each compound in GC and GC×GC have been distinguished and have been labeled in Table 1. In addition, we have checked the entire form to revise and improve the details.